# Nanostructure Engineering via Intramolecular Construction of Carbon Nitride as Efficient Photocatalyst for CO_2_ Reduction

**DOI:** 10.3390/nano11123245

**Published:** 2021-11-29

**Authors:** Muhammad Sohail, Tariq Altalhi, Abdullah G. Al-Sehemi, Taha Abdel Mohaymen Taha, Karam S. El-Nasser, Ahmed A. Al-Ghamdi, Mahnoor Boukhari, Arkom Palamanit, Asif Hayat, Mohammed A. Amin, Wan Izhan Nawawi Bin Wan Ismail

**Affiliations:** 1Yangtze Delta Region Institute (Huzhou), University of Electronic Science and Technology of China, Huzhou 313001, China; sohailncp@gmail.com; 2Department of Chemistry, College of Science, Taif University, P.O. Box 11099, Taif 21944, Saudi Arabia; ta.altalhi@tu.edu.sa; 3Department of Chemistry, Faculty of Science, Research Center for Advanced Materials Science (RCAMS), King Khalid University, P.O. Box 9004, Abha 61413, Saudi Arabia; agsehemi@kku.edu.sa; 4Physics Department, College of Science, Jouf University, Sakaka 75471, Saudi Arabia; themaida@ju.edu.sa; 5Physics and Engineering Mathematics Department, Faculty of Electronic Engineering, Menoufia University, Menouf 32952, Egypt; 6Chemistry Department, College of Science and Arts, Jouf University, Al-Gurayyat 77447, Saudi Arabia; karamsaif@ju.edu.sa; 7Chemistry Department, Faculty of Science, Al-Azhar University, Assiut 71524, Egypt; 8Department of Physics, Faculty of Science, King Abdulaziz University, Jeddah 21589, Saudi Arabia; agamdi@kau.edu.sa; 9College of Material Science & Engineering, Beijing University of Technology, Beijing 100081, China; bukharimahnoor11@gmail.com; 10Energy Technology Program, Department of Specialized Engineering, Faculty of Engineering, Prince of Songkla University, 15 Karnjanavanich Rd., Hat Yai, Songkhla 90110, Thailand; energy_man001@hotmail.com; 11State Key Laboratory of Photocatalysis on Energy and Environment, College of Chemistry, Fuzhou University, Fuzhou 350116, China; 12Faculty of Applied Sciences, Universiti Teknologi MARA, Cawangan Perlis, Arau Perlis 02600, Malaysia

**Keywords:** carbon nitride (CN), dihydroxy benzene (DHB), photocatalysis, copolymerization, CO_2_ reduction

## Abstract

Light-driven heterogeneous photocatalysis has gained great significance for generating solar fuel; the challenging charge separation process and sluggish surface catalytic reactions significantly restrict the progress of solar energy conversion using a semiconductor photocatalyst. Herein, we propose a novel and feasible strategy to incorporate dihydroxy benzene (DHB) as a conjugated monomer within the framework of urea containing CN (CNU-DHBx) to tune the electronic conductivity and charge separation due to the aromaticity of the benzene ring, which acts as an electron-donating species. Systematic characterizations such as SPV, PL, XPS, DRS, and TRPL demonstrated that the incorporation of the DHB monomer greatly enhanced the photocatalytic CO_2_ reduction of CN due to the enhanced charge separation and modulation of the ionic mobility. The significantly enhanced photocatalytic activity of CNU–DHB_15.0_ in comparison with parental CN was 85 µmol/h for CO and 19.92 µmol/h of the H_2_ source. It can be attributed to the electron–hole pair separation and enhance the optical adsorption due to the presence of DHB. Furthermore, this remarkable modification affected the chemical composition, bandgap, and surface area, encouraging the controlled detachment of light-produced photons and making it the ideal choice for CO_2_ photoreduction. Our research findings potentially offer a solution for tuning complex charge separation and catalytic reactions in photocatalysis that could practically lead to the generation of artificial photocatalysts for efficient solar energy into chemical energy conversion.

## 1. Introduction

The production of renewable fuels with rich CO_2_ as raw materials through solar energy has been considered the best solution for energy crises and environmental remediations [1,2,3,4,5]. It is always attracting and has received much attention for generating renewable fuel from the conversion of CO_2_ through semiconductor-based photocatalysts [5,6,7,8]. To date, a variety of semiconductor photocatalysts including CeO_2_, TiO_2_, Ga_2_O_3_, ZnO, ZnGe_2_O_4_, and Bi_2_WO_6_ have been used for the reduction of CO_2_ source [9,10,11,12,13,14,15]. Unfortunately, such semiconductor materials have a broad bandgap that absorbs only ultraviolet light irradiation and has less photocorrosion potential. Furthermore, bulk semiconductors typically have a high recombination rate of photoinduced charge carriers [16,17]. All of these disadvantages substantially jeopardize the effectiveness and successful long-term implications of CO_2_ photoconversion on the basis of photocatalysts. As a result, it is vital for photocatalytic CO_2_ conversion to fabricate and construct visible light-responsive, highly efficient, and durable catalysts [5,18,19]. Numerous strategies, such as morphological tailoring, junction creation, crystal facet engineering, and surface modification, can be used to prevent the recombination of highly reactive photogenerated carriers in photocatalysis. Despite this, the photocatalytic performance is limited due to a lack of active sites and photocorrosion features. The proper coupling of one semiconductor with other materials to form a heterojunction may aid in the separation of spatial charges and protect the light-harvesting semiconductor from photocorrosion [20,21,22]. 

Numerous articles have been published about heterogeneous photocatalysis since 1981, demonstrating the topic’s increasing popularity [23]. The heterogeneous catalyst has a well-defined structure, and the reaction is multi-sited, involving multiple active sites such as edge, face, and defect simultaneously [24]. Photocatalytic materials such as metal oxides, sulfides, and nitrides have been studied in the literature widely [25]. CdS is a semiconductor with a bandgap of 2.42 eV and an absorption peak of 514 nm. Consequently, CdS is more effective at absorbing visible light or UV radiation with a wavelength of less than 514 nm [26]. Aside from water decomposition [26] and CO_2_ reduction [27], CdS semiconductors have a bandgap location that is ideal for several photocatalytic processes. Furthermore, the CdS conduction band edge is lower than the other common semiconductors (such as TiO_2_, SrTiO_3_, and ZnO [28]), which means that photoelectrons of CdS have a stronger reducing power in the photocatalytic reaction. As a result, the photocatalytic properties of CdS have been extensively studied. However, the CdS material is susceptible to light corrosion, which limits the number of photocatalysts that can be restored.

Carbon nitride (CN) has received considerable attention as a metal-free organic semiconductor due to its remarkable thermal and chemical stability and favorable electrical structure [29,30,31,32,33,34,35]. CN has been shown to be a potential solar energy conversion candidate [36,37]. The synthesis of bulk CN by the direct co-condensation process limits their applications because of its low surface area and less active site [13,34,38,39,40]. To improve the quantum efficiency in the visible light region, the above-mentioned drawbacks must be overcome that limit the applications of CN [35,41]. The fabrication of hierarchical micro/nanostructures is an effective method for addressing these issues. Micro/nanostructured CN was obtained using a variety of techniques, including hard templating, which was commonly used to obtain porous, sphere, and tubular carbon nitride (CN) [13,39]. However, the templating technique is expensive, time-intensive, and not eco-friendly, given the environmentally toxic reagents to be used for the extraction of the prototype, which prevents further functionalization. A variety of techniques have been used to modify CN, including doping, morphology tailoring in the form of nanorods, and hollow nano-spheres junction fabrication, crystal facet engineering, and surface modification [5,42,43,44]. Inspired by these advancements, molecular engineering (copolymerization) has arisen as a new significant technique by incorporating new energetic organic conjugated monomers within the framework of CN to improve its photocatalytic properties [5,33,37,39,43,44]. Incorporating these organic motifs into the CN structure via copolymerization enhances photogenerated electron accumulation and transport, leading to improved photocatalytic activity under visible light irradiation [5,33,37]. To achieve high photocatalytic efficiency and stability, this copolymerization grafts organic motifs within the CN skeleton by using a simple one-step condensation technique [42]. From the perspective of this approach, Nie et. al. developed Z-scheme g-C_3_N_4_/ZnO microspheres for CO_2_ reduction [45]. Hasija et al. designed Z-scheme g-C_3_N_4_/AgI/ZnO/CQDs photocatalysts for the efficient photodegradation of 2,4-dinitrophenol, which exhibited remarkable stability and recyclability [21]. Hayat et al. examined the introduction of trimesic acid as a conjugated co-monomer within the CN framework that was highly effective in photocatalytic CO_2_ reduction [5,18]. Intramolecular conjugating monomers are of great interest and are discovered as energized candidates for intramolecular donor–acceptor behaviors within the triazine subunit of CN to boost its photocatalytic activity [42]. CN has been employed for CO_2_ photoreduction by several groups due to its large surface area and distinct semiconductor properties [46,47]. These features may lead to the formation of charge carriers capable of activating CO_2_ molecules at the active sites of the nitrogen-rich CN skeleton. Liu et al. demonstrated the synthesis of layered heterojunction photocatalysts (PCN/ZnIn_2_S_4_) via the in situ growth of 2D ZnIn_2_S_4_ nanosheets on the surfaces of ultrathin CN layers for improved CO_2_ conversion under visible light [48]. In addition, the CN/ZnIn_2_S_4_ composites also demonstrate enhanced photoactivity for deoxygenated CO_2_ conversion. On the other hand, Xue et al. [49] proposed a new strategy for solar fuel production with simultaneous organic synthesis using photo-holes oxidation power on amphiphilic metal-free semiconductors. The successfully grafted pyrene functional group onto the CN surface via a post copolymerization technique (Py-CN) shows unique biphasic photocatalytic activities that allow efficient CO_2_ photoreduction in aqueous solution while effectively oxidizing alkenes (C=C) in the organic phase. This is due to the pyrene functional group enhanced lipophilicity, allowing hydrophobic alkene molecules to reach the CN surface and react with hydroxyl radicals (OH) produced by photogenerated holes [50].

In the present work, we use urea containing CN to introduce hydroxyl groups containing benzene rings, i.e., 1,4-dihydroxybenzene organic conjugated monomer (DHB) through copolymerization and studied the versatile application of products for photocatalytic CO_2_ reduction. The incorporation of an aromatic benzene ring into the triazine analogous of CN can produce a regular inner structure of CN as a spinal candidate to achieve a polymeric conjugated C–N network system. The aromaticity of the benzene ring in the CN framework can improve the photo-excitation process of electrons from the ground state to the excited state, as well as photogenerated charge separation. As discussed above, the integrated monomer into the CN skeleton will be either an electron donor or an acceptor; therefore, the dihydroxybenzene monomer acts as an electron donor. Figure 1 provides a proposed reaction mechanism for the incorporation of dihydroxybenzene (DHB) into the CN framework by donating electrons to the pristine CN to make a long polymer network that has high stability and a large surface area. An as-synthesized sample showed good photocatalytic activity of CO_2_ reduction into CO and H_2_ sources, which was roughly 4-fold higher than the parental CN. This effort will open up new possibilities to promote the use of copolymerized CN for CO_2_ photofixation under visible illumination.

## 2. Experimental

### Synthesis of CN and Copolymerized CN Photocatalysts

A pristine CN sample was prepared by taking a specific amount of urea as a precursor in a 30 cm × 60 cm rectangle crucible dish, and the resulting product was annealed at 550 °C in an air furnace for 2 h at a heating rate of 4.5 °C min^−1^. After calcinations, the yellow color sample is obtained and labeled as CNU (U denotes the urea containing carbon nitride). Similarly, the copolymerized samples were synthesized by taking 10 g of urea with varied amounts of 1,4-dihydroxybenzene (DHB) monomer in 15 mL distilled in an oil bath system by heating at 100 °C through vigorous stirring to remove water thoroughly. After the evaporation of water, the solid samples were transferred into a 30 cm × 60 cm rectangle crucible one by one and heated for 2 h at 550 °C at a 4.5 °C min^−1^ rate. After heating, various color-containing samples were obtained depending on the amount of DHB monomer to be copolymerized. The as-prepared samples were marked as CNU-DHBx where x determines different amounts of DHB to be copolymerized with carbon nitride (x = 0.0050, 0.0100, 0.0150, 0.0200, 0.0250, 0.0300 g), respectively. Up to a specific amount of DHB with CNU, the color of the samples was obtained from dark yellow to brown. All prepared samples were characterized for different techniques without any further purification or washing.

## 3. Result and Discussion

As-synthesized samples of pristine and copolymerized CN samples were characterized by employing the X-ray diffraction (XRD) technique [39,51,52]. Figure 2a demonstrates that there are no extra peaks observed in the XRD of pristine CNU and copolymerized CNU-DHBx, respectively. All samples had two distinct peaks, and the evidence indicated the same structural composition; thus, no detection of extra impurity peaks was observed. The peak at 12.9° indexed as (1 0 0) for all samples represents the inter-layer distance in the structural repeating unit of triazine motifs. The distance between these repeating units is approximately 0.68 nm. A dominant peak at 27.7° (0 0 2) is due to the periodic interlayer stacking of the conjugated aromatic system of CN at 0.32 nm. This periodic stacking distribution in the conjugated system of CN could be determined from the integration of DHB monomer within the skeleton of CNU. For as-synthesized samples, FTIR spectroscopy was used to determine the chemical composition of all samples [39,53]. Figure 2b illustrates the stretching and bending vibration of the heptazine ring unit, which is assigned from several peak locations of the absorption bands at 810 cm^−1^ and 1200–1630 cm^−1^, respectively. All of the prepared samples have one common broad peak, which is located between 3000 and 3600 cm^−1^ and attributed to the stretching modes of the O–H and N–H groups, which confirms the presence of the amino and hydroxyl group that originated from the condensation process. Another peak located at 1200 to 1600 cm^−1^, which is due to the breathing mode of 810 cm^−1^ triazine units, is quite similar to that of CNU. The XRD and FTIR analysis demonstrate that pristine CNU and copolymerized CNU-DHBx have similar a phase composition and chemical structure. Thus, the incorporation of a DHB monomer in the framework of CNU does not carry any change in the crystal and chemical integrity structure of CNU [39,42].

Similarly, the induction of conjugated DHB within the skeleton of CN alters several changes in the chemical analysis of CN that can be confirmed from solid-state ^13^C NMR spectra and XPS analysis, respectively. The XPS survey spectra were investigated to figure out the chemical composition states within samples. Figure 3a,b demonstrate the XPS wide spectrum for parental CNU and copolymerized CNU-DHB_15.0_ samples mainly composed of carbon (C), nitrogen (N), and oxygen (O), respectively. The O1 s peak detected at 533 eV for both CNU and CNU-DHBx arises due to the adsorption of atmospheric moisture during synthesis. High-resolution XPS spectra of C 1s for CNU-DHB_15.0_ and CNU composed of two distinct peaks are altered; a small shifting occurs in the perception of peaks indexed at 284.3 eV for CNU and 284 eV for CNU-DHB_15.0_, which is attributed to the sp^2^ C–C bonds (Figure 3c,d). However, the other peak indexed at 287.5 eV for CNU and at 288 eV for CNU-DHB_15.0_ ascribed the sp^2^ hybridized carbon located in the cage of the N-containing aromatic ring (N–C=N). Furthermore, the high-resolution XPS spectra of N 1s also alters the shifting of peaks de-convoluted into four peaks. The main peak centering at 398 eV for CNU and 398.7 eV for CNU-DHB_15.0_ corresponds to the sp^2^ hybridized nitrogen within the carbon (C−N=C). Hence, the other peaks indexed at 398.8, 400.3, and 403.8 eV for CNU and 399.7, 401.2, and 404.3 eV for CNU-DHB_15.0_ display the bridging tertiary nitrogen (N–(C)_3_), graphitic N, and amino functional groups (C−N−H) [39,42], as illustrated in Figure 3e,f.

The solid-state ^13^C NMR spectra for CNU and CNU-DHB_15.0_ samples were used to examine the extra evolution of carbon contents after the copolymerization process (Appendix A). All of the peak alignments in the NMR spectra for both samples are the same, but a new peak was indexed at 135.7 ppm for the CNU-DHB_15.0_ sample, thus confirming the increased amount of carbon. The morphology and microstructure of pure CNU and CNU-DHB_15.0_ were analyzed by FESEM and TEM, as illustrated in Figure 4. The FESEM morphology of pure CNU obviously demonstrates an agglomerated shape having irregular small stacking flakiness, as displayed in Figure 4a,b, while this morphology became elongated in size after the inducing of a DHB monomer within CNU, having a large cloudy superficial area, as shown in Figure 4c,d. The typical TEM images of pure CNU (Figure 4e,f) and CNU-DHB_15.0_ (Figure 4g,h) exhibit platelet ribbon-like distorted morphology for pure CNU samples. After copolymerization, the surface morphology become dense, stacked, and proliferous, which results in an increase in the surface area.

The UV-Vis diffuse reflectance spectroscopy (DRS) of the samples was used to assess the effect of the DHB monomer on the optical property of CNU, as shown in Figure 5a. After DHB incorporation into the backbone of CNU, the optical bandgap is significantly reduced, which facilitates the harvesting of light and photogenerated carriers [39,42]. To study the transfer and exciton separation behavior of the photogenerated electrons and holes of the as-prepared copolymerized samples, we carried out photoluminescence (PL) spectra at room temperature under 370 nm excitation (Figure 5b). All these samples show a wide broad peak indexed at around 455 nm and extend their tail to 600 nm. The spectra demonstrate two types of band: i.e., a shoulder band, which is attributed to the emission process of CN from the valence band (LUMO) toward the conduction band (HOMO) at a shorter wavelength of 440 nm. The other PL band corresponds to the emission of charges as found in donor–acceptor polymers [54]. Actually, in CNU, the weak shoulder band is produced due to the emission of charges at about 463 nm. Similarly, the rates of transfer of photogenerated charges toward transition states are much higher in copolymerized samples compared to pristine samples. To understand the exciton separation behavior, we conducted surface photovoltage. When compared to the copolymerized sample, pristine CNU exhibits a weak SPV signal at 350–550 nm (Figure 5c), which is due to the low charge separation compared to pristine. Nonetheless, the copolymerized CNU exhibits a clear enhancement of the SPV signal, indicating an increased charge separation due to the aromaticity of the benzene ring in the CN framework, which acts as an electron-donating species. The effect of the aromaticity of the benzene ring of DHB in the framework of CN, acting as an electron-donating species, was further verified by OCP spectra, as shown in Figure 5d. The C/N ratio and bandgap for all of the as-synthesized samples are depicted in Appendix A, in which the superior sample CNU-DHB_15.0_ demonstrates a boosted C/N ratio and low bandgap, respectively. 

The Brunauer–Emmett–Teller (BET) method was conducted to examine the specific surface area of as-synthesized samples (Figure 6a). Both samples (CNU and CNU-DHB_15.0_) represent the N_2_ adsorption–desorption isotherms having H_3_ hysteresis-type loops [55]. The surface area of blank CNU is 49.9 m^2^/g and after the copolymerization process, the surface area of CNU-DHB_15.0_ improved to 123.7 m^2^/g, respectively. Actually, large surface areas supply an abundance of energized sites for the photocatalytic reaction that result in photocatalytic performance. Figure 6b illustrates the BJH pore size distribution and highlights that the catalysts’ pore size decreases and pore volume increases, hence originating the structure of samples as mesoporous. Actually, during the co-condensation process of samples, various gases evolved from their partial decomposition and adopted mesoporous structures [39,42]. The CO_2_ adsorption isotherm was evaluated for pristine CNU and copolymerized CNU-DHB_15.0_, as illustrated in Figure 6c, indicating that CNU-DHB_15.0_ has remarkable CO_2_ adsorption ability compared to pure CNU. The surface area for all of the as-synthesized samples was depicted in Appendix A, in which the superior sample CNU-DHB_15.0_ demonstrates a high surface area.

The photocatalytic activity of CO_2_ reduction was carried out under visible light illumination (λ > 420 nm). Typically, 30 mg of photocatalyst was dispersed in a solvent (MeCN)/H_2_O = 5:1) containing Co(bpy)_3_Cl_2_·6H_2_O as a photosensitizer and triethanolamine (TEOA) as a sacrificial electron donor [18]. The reactions were carried out for half an hour to five hours, generating a significant amount of CO (32.5 µmol/h^−1^) in the first hour and achieving a high CO efficiency, i.e., 85 µmol/h in five hours (Figure 7a). Similarly, bulk CNU photocatalysts have poor photocatalytic performance compared to CNU-DHB_15.0_. The photocatalytic reduction of CO_2_ was enhanced after the incorporation of DHB in the framework of CNU, indicating the best activity in optimal condition (Figure 7b). All of the other copolymerized samples demonstrate good photocatalytic performance and are much better than pristine CNU. In all of these photocatalysts, CNU-DHB_15.0_ manifested a boosted performance for the photocatalytic CO_2_ reduction under visible light (Appendix A). Similarly, the activity of samples decreases due to the appearance of a negative site from the excess amount of DHB monomer that destroys the conjugated system of CN. Figure 7c demonstrates the recycle stability of the CNU-DHB_15.0_ photocatalyst, which depicts a high photocatalytic stability toward CO_2_ reduction in every phase. The long durability experiments demonstrate a decline in the few cycles, which is correlated with the erosion of the co-catalyst cobalt due to the long interaction of solar light during a reaction.

Long-range wavelength experiments have been conducted for the superior sample CNU-DHB_15.0_ to investigate the effect of light source on the photocatalytic products (Figure 7d). The results manifest that the light of a longer wavelength decreases the photocatalytic evolution of CO and H_2_, suggesting that light-harvesting phenomena is responsible for the CO_2_ photoreduction by generating stimulated electrons [18]. The EPR and life time spectra of as-synthesized samples were conducted, as demonstrated in Figure 7e,f. The life time decay of superior sample is much enhance than the blank samples and same the EPR peak intensity of the CNU-DHB_15.0_ sample improved and broadened immediately as compared to pristine CNU. Upon copolymerization, the unpaired electrons of carbon atoms increase the EPR peak intensity and also increase the delocalization of π-conjugated clusters. Most photochemical radical pairs generate light on the catalyst surface by trapping it under visible light [56,57,58]. A comparison of the reported results of the photocatalytic CO_2_ reduction with our current research work is presented in Appendix A.

Similarly, several isotopic controlled experiments were held using ^13^CO_2_ as feedback to analyze the insights of carbon contents of the produced carbon monoxide (CO) products under similar conditions. After irradiation of one hour, the gas chromatography and mass spectrometry (GC-MS) demonstrate that the peak produced at 2.4 mint having *m*/*z* = 29 originated due to the reactant of ^13^CO_2_, as illustrated in Figure 8a,b. The result strongly suggests that photocatalytic CO_2_ reduction into the CO product is absolutely performed by using the original reactant CO_2_ gas.

The photocatalytic activities of the synthesized samples were evaluated for the degradation of rhodamine B under visible light irradiation (λ = 420 nm), as shown in Appendix A. The CNU-DHBx catalyst shows better photocatalytic activities than pure CNU. The degradation rate of rhodamine B is enhanced after the copolymerization process. Similarly, few experiments for the degradation of RhB were conducted under dark illumination, and no noticeable photodegradation activity was observed, so it was concluded that the reaction and catalysts are only active under visible light illumination. The overall degradation was carried out for 70 min under visible light at varying exposure times. The kinetic of RhB degradation using the CNU-DHB_15.0_ photocatalyst (Appendix A) was performed successfully and was calculated with a pseudo-first-order equation as fellows.
ln (C_0_/C) = kt(1)
where the apparent rate constant of the pseudo-first order is shown in the above Equation (1) k, the time of irradiation applied during the degradation process shall be t, and the initial and final concentration of the RhB solution shall be C_0_/C [16]. The result confirms that CNU-DHB_15.0_ has a better kinetic rate constant value and also investigates the pseudo-first-order catalytic rate constants from the slope of the plots, which are four times higher than that of pure CNU. The apparent rate constants for RhB degradation through different synthesized samples are depicted in Appendix A. The recycling experiments were conducted for the superior photocatalyst CNU-DHB_15.0_ in order to check its stability toward the photodegradation of RhB. Appendix A shows the cyclic stability experiments for the RhB, and for this function, the used CNU-DHB_15.0_ catalysts were centrifuged, washed several times with ethanol and water, and dried to reuse in a fresh reaction. The results reveal that CNU-DHB_15.0_ showed good photocatalytic activity and stability in practically all of the four cycling runs, and hence, no obvious decline was observed after long-term use. It indicates that CNU-DHBx is highly stable and can be reused for the treatment of RhB. 

## 4. Photocatalytic Mechanism

The possible scheme for the photocatalytic H_2_ production and CO_2_ with the CNU-DHB_15.0_ photocatalyst was evaluated, as shown in Figure 9. When irritated under solar light, the electrons are excited to the conduction band (CB), leaving positive holes in the valence band (VB) of the CNU-DHB_15.0_ photocatalyst. The excited electrons in the CB of CNU-DHB_15.0_ reduce protons to reduce CO_2_ into the CO source, while the positive holes in the VB contribute to oxidizing TEOA. The particles of cobalt (CO) play an important role in the separation of exited charges through its metallic character called surface catalysis. In the reaction system, the small addition of catalysts with solvent creates splits on the interface of material; then, H^+^ ions are produced, which trap the electrons and generate H_2_ fuel. Although a large number of the photoelectrons and holes produced are attributed to the absorption of solar photons, a very small portion of these charges is used for effective photocatalysis. To accelerate the photocatalytic production of H_2_ fuel, the separation of electrons and holes through CO and sacrificial agent TEOA is very important. Similarly, the induced holes in the VB of the superior sample results are oxidized, which participate in the photocatalytic degradation of RhB reduction under visible light illumination (λ = 420 nm).

## 5. Conclusions

The modification of carbon nitride (CNU) for photocatalytic CO_2_ reduction is an enticing research topic owing to the growing severity of fuel and ecological ailments. In this study, the organic aromatic co-monomer dihydroxybenzene (DHB) was thermally incorporated within carbon nitride (CNU referred to as urea-based carbon nitride) through the copolymerization process recognized as CNU-DHB. Interestingly, the copolymerized samples demonstrated an efficient CO_2_ reduction due to the aromaticity conjugated π electrons of the benzene ring in the framework of CN, which acts as an electron-donating species; therefore, it can speed up the process of photogenerated charge separation. This assimilation altered a significant change in the electronic structure of CNU, boosted its electron transport, and increased the photocatalytic properties of CNU under visible light irradiation. The CNU–DHB_15.0_ catalyst yielded 85 μmol/h of CO and 19.92 µmol/h of H_2_ source after 5 h of irradiation, highlighting the maximum yield of photocatalytic performance that is almost four times higher than that of parental CNU. Such an approach predicts a substantial distraction in the precise surface area, energy gap, and chemical properties, and it promotes the effective segregation of photoinduced load carriers from HUMO toward LOMO of CNU, making it an ideal alternative for photocatalytic CO_2_ reduction reactions. Thus, the CNU-DHB_15.0_ composite photocatalyst provides a useful guide for the synthesis of efficient photocatalysts for photocatalytic applications.

## Figures and Tables

**Figure 1 nanomaterials-11-03245-f001:**
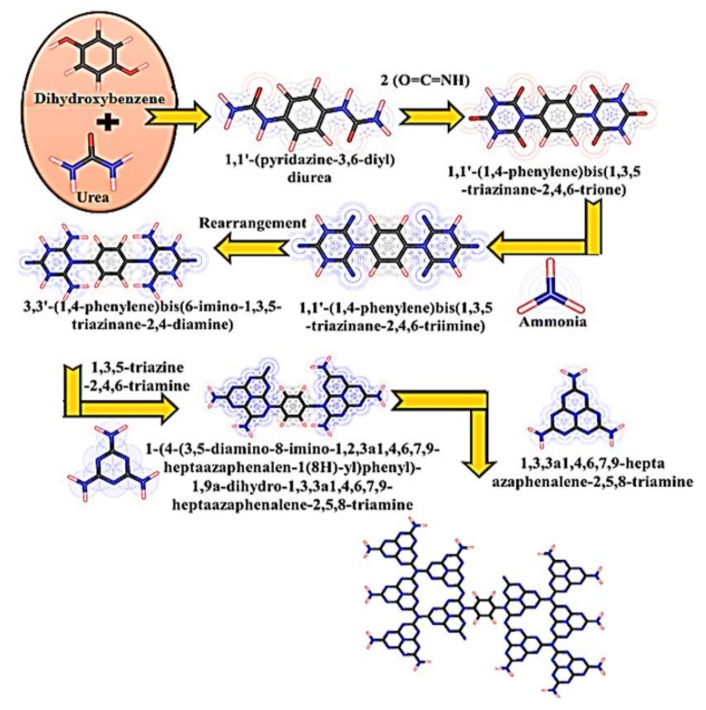
Proposed reaction mechanism of incorporating dihydroxybenzene (DHB) into CN networks.

**Figure 2 nanomaterials-11-03245-f002:**
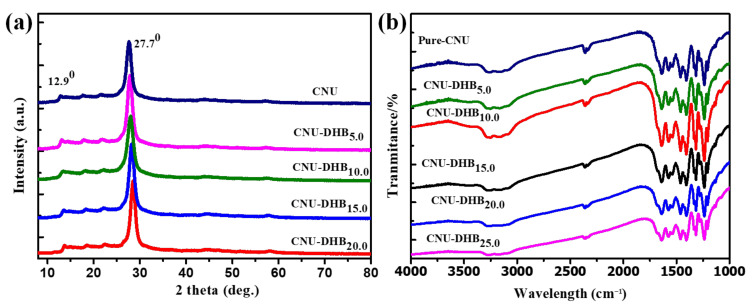
(**a**) XRD spectrum and (**b**) FTIR spectra of CNU and CNU–DHBx samples.

**Figure 3 nanomaterials-11-03245-f003:**
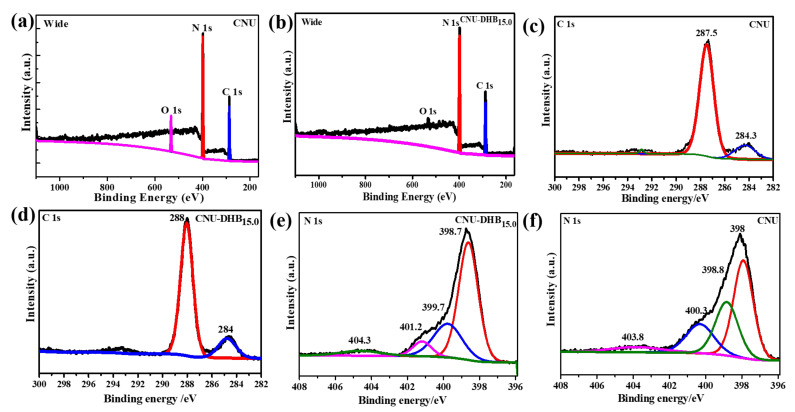
(**a**,**b**) Wide spectrum, (**c**,**d**) C 1s, and (**e**,**f**) N 1s high-resolution XPS spectra of the CNU–DHB_15.0_ and CNU samples.

**Figure 4 nanomaterials-11-03245-f004:**
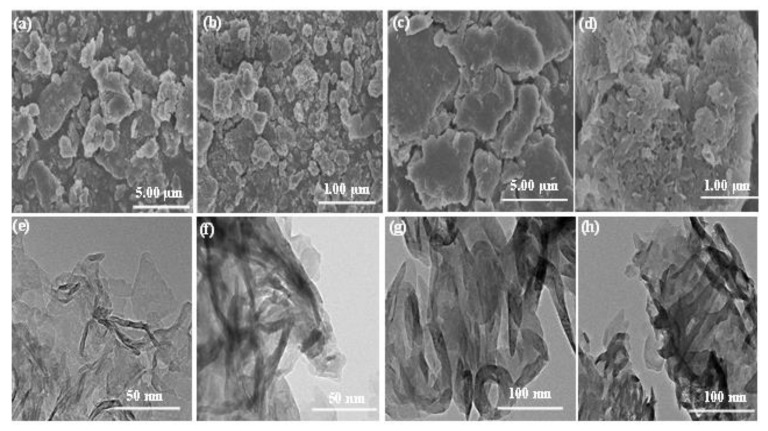
FESEM of CNU (**a**,**b**) and CNU–DHB_15.0_ (**c**,**d**), TEM images of CNU (**e**,**f**) and CNU–DHB_15.0_ (**g**,**h**).

**Figure 5 nanomaterials-11-03245-f005:**
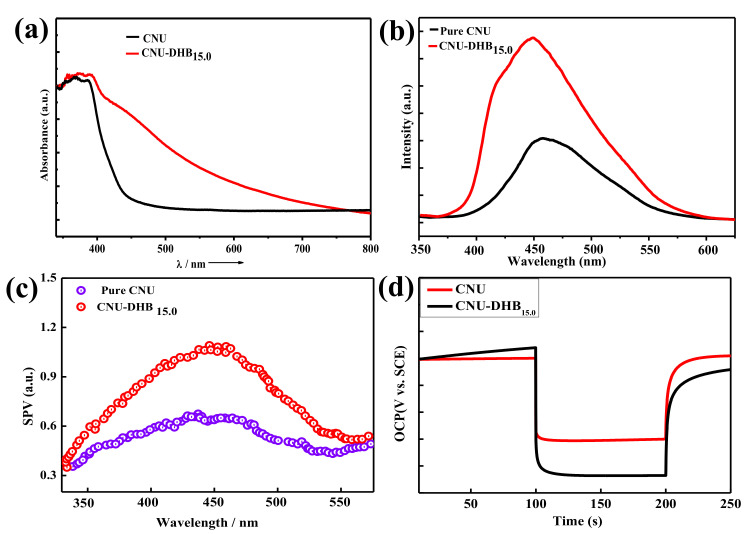
(**a**) UV-Vis DRS spectra of the CNU and CNU–DHBx samples under 370 nm excitation. (**b**) The photoluminescence spectrum of the CNU and CNU–DHBx samples at room temperature. (**c**) SPV spectra of the CNU and CNU–DHBx samples. (**d**) OCP response curve of the CNU and CNU–DHBx samples.

**Figure 6 nanomaterials-11-03245-f006:**
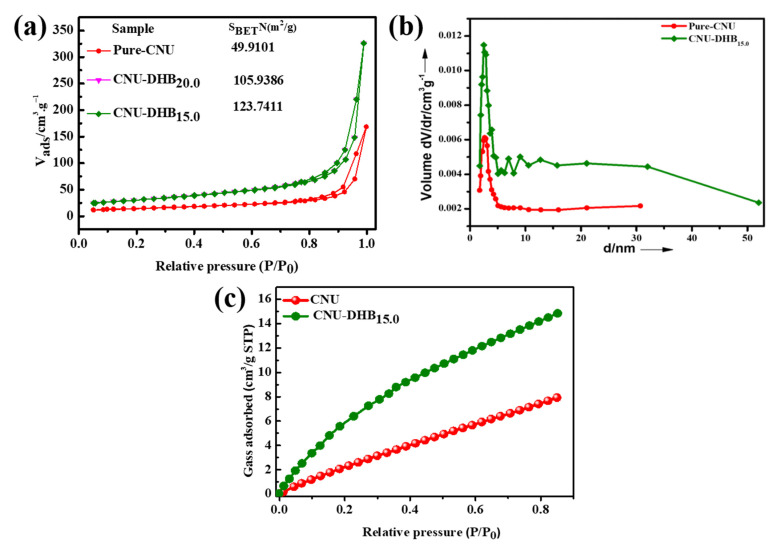
(**a**) N_2_ adsorption–desorption isotherms (77 K), (**b**) BJH pore size distribution, and (**c**) CO_2_ adsorption isotherms (273 K) for CNU and CNU-DHB_15.0_ samples.

**Figure 7 nanomaterials-11-03245-f007:**
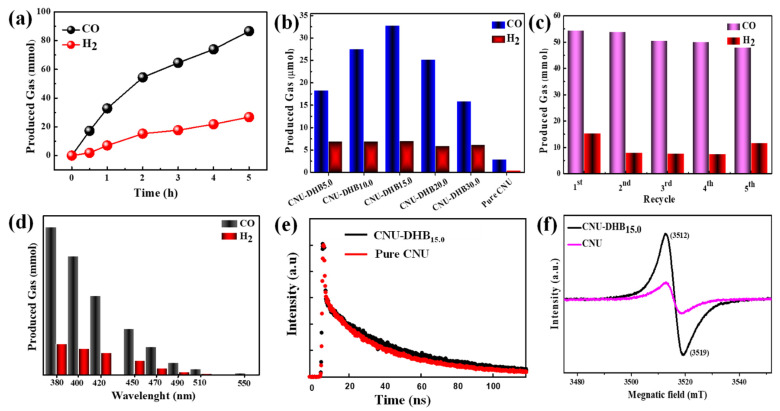
Reduction (**a**) Time-production plot between CNU and CNU-DHB_15.0_, (**b**) Comparison of photoreduction with different photocatalysts synthesized, (**c**) Recycling stability test, (**d**) Different wavelength experiments, (**e**) TRPL spectra of CNU-DHB_15.0_ and pure CNU, (**f**) EPR spectra of CNU and CNU-DHB_15.0_.

**Figure 8 nanomaterials-11-03245-f008:**
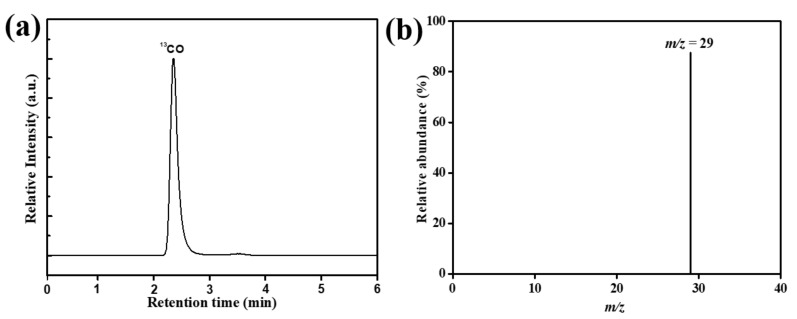
(**a**) Gas chromatogram and (**b**) mass spectrum analysis of the carbon products using CNU-DHB_15.0_ as catalyst and ^13^CO_2_ as reactant.

**Figure 9 nanomaterials-11-03245-f009:**
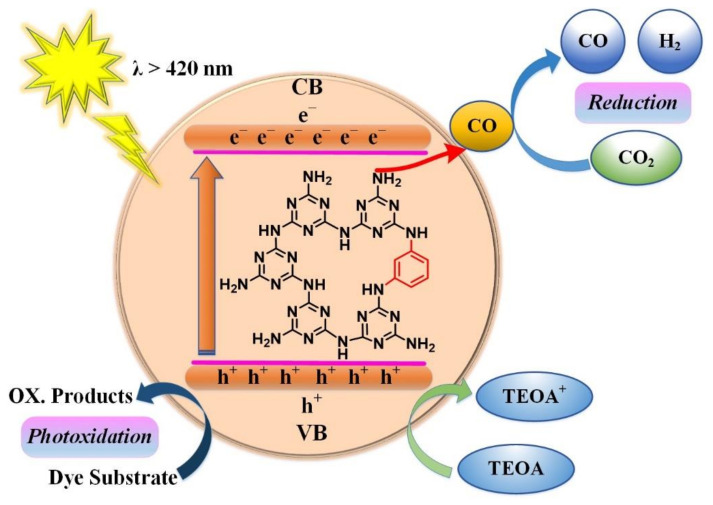
The proposed reaction mechanism for photo reduction and degradation through CNU-DHB_15.0_.

## Data Availability

The data presented in this study are available on request from the corresponding author.

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
