# Peer review of "Nanostructure Engineering via Intramolecular Construction of Carbon Nitride as Efficient Photocatalyst for CO2 Reduction"

_nanomaterials, 2021, doi:10.3390/nano11123245_

Round 1

Reviewer 1 Report

The article entitled as Nanostructure Engineering via Intramolecular construction of 2 carbon nitride as efficient photocatalyst for CO2 Reduction has scientific value and relevance.   It is an timely manuscript. However, there are some gap in this. The following point must be considered before final submission  

  1. As for as semiconductor photocatalyst are concerned, adsorption plays very critical role.  The authors must add a paragraph involving heterogeneous photocatalysis. The authors must discuss basics of heterogeneous photo catalytic process by citing the. Latest work.  

https://doi.org/10.1016/S1010-6030(97)00118-4

https://doi.org/10.1016/B978-0-12-813926-4.00028-8

https://doi.org/10.1039/C9CS00172G

https://doi.org/10.1039/C9CS00172G

doi:10.3390/app9122489

  1. The authors must mention the advantages of photocatalysis over other treatment techniques.  Plz discuss with reference to pioneer work.

https://doi.org/10.1016/j.jece.2019.103272.

    http://nopr.niscair.res.in/handle/123456789/5300

  1. The heterojunction formation is another important strategy for enhancing photocatalytic activity. Also, Z scheme is regarded as one of the best way to improve photocatalytic activity. Plz elaborate with help of following work.

    https://doi.org/10.1016/j.jece.2020.104483.

    https://doi.org/10.1016/j.ces.2020.116219

     https://doi.org/10.1016/j.jtice.2020.08.003.

  1. How the reusability photocatalyst was assured?
  2. The conclusion must be re-written in more concise way.
  3. The mechanism figure was very crowded
  4. The GA should be improved.

Author Response

Comments and Suggestions for Authors

The article entitled as Nanostructure Engineering via Intramolecular construction of 2 carbon nitride as efficient photocatalyst for CO2 Reduction has scientific value and relevance.   It is a timely manuscript. However, there are some gap in this. The following point must be considered before final submission  

Q.1. As for as semiconductor photocatalyst are concerned, adsorption plays very critical role.  The authors must add a paragraph involving heterogeneous photocatalysis. The authors must discuss basics of heterogeneous photo catalytic process by citing the. Latest work.  

Reply. Thanks for your experience comments. Dear Professor, the suggested research articles are cited in the introduction section of the revised manuscript. The details are as follows.

  1. Mills, A. and S. Le Hunte, An overview of semiconductor photocatalysis. Journal of photochemistry and photobiology A: Chemistry, 1997. 108(1): p. 1-35.
  2. Widegren, J.A. and R.G. Finke, A review of the problem of distinguishing true homogeneous catalysis from soluble or other metal-particle heterogeneous catalysis under reducing conditions. Journal of Molecular Catalysis A: Chemical, 2003. 198(1-2): p. 317-341.
  3. Belver, C., et al., Semiconductor photocatalysis for water purification, in Nanoscale Materials in Water Purification. 2019, Elsevier. p. 581-651.
  4. Zhang, F., et al., Recent advances and applications of semiconductor photocatalytic technology. Applied Sciences, 2019. 9(12): p. 2489.
  5. Zhang, L., et al., Characterization of semiconductor photocatalysts. Chemical Society Reviews, 2019. 48(20): p. 5184-5206.
  6. Xu, Y. and M.A. Schoonen, The absolute energy positions of conduction and valence bands of selected semiconducting minerals. American Mineralogist, 2000. 85(3-4): p. 543-556.

Q.2. The authors must mention the advantages of photocatalysis over other treatment techniques.  Plz discuss with reference to pioneer work.

Reply. Thanks for your experience comments. Dear Professor, the suggested research articles are cited in the introduction section of the revised manuscript. The details are as follows

  1. Hasija, V., et al., Carbon quantum dots supported AgI/ZnO/phosphorus doped graphitic carbon nitride as Z-scheme photocatalyst for efficient photodegradation of 2, 4-dinitrophenol. Journal of Environmental Chemical Engineering, 2019. 7(4): p. 103272.
  2. Pare, B., P. Singh, and S. Jonnalgadda, Degradation and mineralization of victoria blue B dye in a slurry photoreactor using advanced oxidation process. 2009.

Q.3. The heterojunction formation is another important strategy for enhancing photocatalytic activity. Also, Z scheme is regarded as one of the best ways to improve photocatalytic activity. Plz elaborate with help of following work.

Reply. Reply. Thanks for your experience comments. Dear Professor, the suggested research articles are cited in the introduction section of the revised manuscript. The details are as follows.

  1. Kumar, A., et al., An overview on polymeric carbon nitride assisted photocatalytic CO2 reduction: strategically manoeuvring solar to fuel conversion efficiency. Chemical Engineering Science, 2021. 230: p. 116219.
  2. Hasija, V., et al., Carbon quantum dots supported AgI/ZnO/phosphorus doped graphitic carbon nitride as Z-scheme photocatalyst for efficient photodegradation of 2, 4-dinitrophenol. Journal of Environmental Chemical Engineering, 2019. 7(4): p. 103272.
  3. Sudhaik, A., et al., Synergistic photocatalytic mitigation of imidacloprid pesticide and antibacterial activity using carbon nanotube decorated phosphorus doped graphitic carbon nitride photocatalyst. Journal of the Taiwan Institute of Chemical Engineers, 2020. 113: p. 142-154.

Q.4. How the reusability photocatalyst was assured?

Reply. Thanks for your experience comments. Dear Professor, we conducted stability (reusability) experiments by applying long photocatalytic rate for several phases in order to see the stability of superior catalyst. After each phase, we only inject the photogenerated gases into the GC and then again start the irradiation after each phase injection. Result demonstrates that our superior sample depicting a persistence in all photocatalytic performance. We already discuss the reusability experiments in our result and discussion text and I am going to height light as yellow for your convenience. Please have a look. Thank you again.

Q.5. The conclusion must be re-written in more concise way.

Reply. Thanks for your experience comments. Dear Professor, we rewrite the conclusion in revised manuscript in more concise way. The details are as follows.

The modification of carbon nitride (CNU) for photocatalytic CO2 reduction is an enticing research topic owing to the growing severity of fuel and ecological ailments. In this study, organic aromatic co-monomer dihydroxybenzene (DHB) was thermally incorporated within carbon nitride (CNU referred as Urea based carbon nitride) through copolymerization process and recognized as CNU-DHB. Interestingly, the co-polymerized samples demonstrated an efficient CO2 reduction due to the aromaticity conjugated π electrons of benzene ring in the framework of CN which act as electron donating species therefore can speed up the process of photogenerated charge separation. This assimilation altered a significant change in the electronic structure of CNU, boosted its electron transport and increased the photocatalytic properties of CNU under visible light irradiation. The CNU–DHB15.0 catalyst yielded 85 μmol/h of CO and 19.92 µmol/h of H2 source after 5 h of irradiation, highlighting the maximum yield of photocatalytic performance that is almost 10 times higher than that of parental CNU. Such approach predicts a substantial distraction in the precise surface area, energy gap, chemical properties, and promotes the effective segregation of photoinduced load carriers from HUMO towards LOMO of CNU, making it ideal alternative for photocatalytic CO2 reduction reaction. Thus CNU-DHB15.0 composite photocatalyst, providing a useful guide for the synthesis of efficient photocatalysts for photocatalytic applications.

Q.6. The mechanism figure was very crowded.

Reply. Thanks for your experience comments. Dear Professor, we properly described mechanism in revised manuscript. The details are as follows.

The possible scheme for the photocatalytic H2 production and CO2 with CNU-DHB15.0 photocatalyst was evaluated, as shown in (Fig. 9). When irritated under solar light, the electrons are excited to the conduction band (CB) leaving positive holes in the valence band (VB) of the CNU-DHB15.0 photocatalyst. The excited electrons in the CB of CNU-DHB15.0 reduce protons to reduce CO2 into CO source, while the positive holes in the VB contribute to oxidizing TEOA. The particles of cobalt (CO) play an important role in the separation of exited charges through its metallic character called surface catalysis. In reaction system, the small addition of water with solvent, splits on the interface of material, H+ ions are produced, which trap the electrons and generate H2 fuel. Although, a large number of photoelectrons and holes are produced attributed to the absorption of solar photons, yet a very small portion of these charges is used for effective photocatalysis. To accelerate the photocatalytic production of H2 fuel, the separation of electrons and holes through CO and sacrificial agent TEOA is very important. Similarly, the induced holes in the VB of superior sample results to oxidized, which participate in the photocatalytic degradation of RhB reduction under visible light illumination (λ = 420 nm)

Q.7. The GA should be improved.

Reply. Thanks for your experience comments. Dear Professor, we improved quality of GA in revised manuscript.

Reviewer 2 Report

The authors present a study on the incorporation of dihydroxy benzene (DHB) into urea-containing carbon nitride (CNU).  The authors varied the DHB content in CNU and assessed their characteristics and performance for CO2 reduction and Rhodamine dye degradation.  The manuscript focused primarily on the characterization of the materials and their performance in photocatalytic CO2 reduction, while the supplementary data contained most of the Rhodamine dye degradation data. 

Generally, I found the manuscript challenging to read.  I recommend that the manuscript be edited to improve readability.  Make sentences concise and make paragraphs shorter.  Each paragraph should have a leading statement followed by supporting statements.  Make sure that all acronyms are clearly defined. 

The authors characterized the materials very well, which makes the study publishable with major revisions.  The revisions that I would like to see in this manuscript include more comprehensive discussions on the material characterizations, which I felt were often lacking in the manuscript.  Describe how data from the characterization methods are used and how they lead to or relate to the photocatalytic activity of the materials.  For many figures, the data was presented but not discussed in sufficient detail to allow the reader to see connections between the characterizations and the photocatalyst performance.  

  • The XRD and FTIR spectra are not impacted at all by the incorporation of DHB?  Does it make sense that there are no observable crystalline  or chemical changes in CN when DHB is incorporated into its structure?  Explain.  Maybe using Figure 1 could explain the results?   
  • The BET adsorption isotherm for the materials is a type 3 adsorption isotherm, which suggests that the adsorbing gas (N2 in this case) has a stronger adherence to itself than with the photocatalyst.   Since one step of the photocatalytic mechanism is adsorption of CO2 onto the surface of the catalyst, I wonder if the adsorption of CO2 could be a limiting step in the photocatalytic reduction of CO2.  
  • Why is the BET surface area of the composite material a factor of 2 or more higher than that of the parent compound?   Explain the possible reasons why DHB would open up surface areas of the materials without impacting the crystalline structure of chemical structure according to XRD and FTIR.  The authors do point out that various gases are liberated from the materials during the synthesis, which may increase mesoporosity.   
  • The hysteresis in this case, as pointed out by the authors, is an H3 type of hysteresis, suggesting that pore structure is irregular and that the pores are  characteristic of flake-like granular materials.  This is supported by the SEM pictures presented in the manuscript.   
  • The diffuse reflectance UV-Vis data, as mentioned by the authors, show that the DHB-CNU materials absorb light in the visible light spectrum, and therefore, they have lower bandgaps than the CNU material by itself.  Can the authors use this data to determine the bandgaps of the materials and report them in this manuscript?  If so, then please do.  If not, then why not? 
  • Explain why the open circuit photo response behavior of  copolymerized samples were inconsistent with the SPV and PL data.  What does this inconsistency mean?  Was the data reproduced to verify it?  

Can you hypothesize why there is an optimum amount of DHB in the composite material, as shown in Figure 7b, using the photocatalytic mechanism as a basis?

In the time-resolved photoluminescence data (7e), it appears that the exciton lifetime of the DHB-CNU composite is shorter than that of the CNU by itself.  This seems to suggest that recombination of electron-hole pairs would be more significant in the composite than in the CNU by itself.  Is this data contradictory of other characterization data?  Why or why not?  

When the authors state that the photocatalyst is efficient in photoreduction of CO2, please define efficiency in the context of the manuscript.  

Table S3 summarizes many efforts in photocatalytic CO2 reduction by incorporating various monomers in urea CN, and so this effort is a variation on a theme of research efforts conducted by one or more of the authors of this manuscript.  Therefore, I would not use the word "novel", since this idea has been published multiple times using other monomers incorporated into CNU. The DHB incorporated into CNU in this study, however, apparently has relatively high photoactivity for CO2 reduction compared to many of the other materials tried by this research group, according to Table S3.  

Author Response

Reveiwer:3.

The authors present a study on the incorporation of dihydroxy benzene (DHB) into urea-containing carbon nitride (CNU).  The authors varied the DHB content in CNU and assessed their characteristics and performance for CO2 reduction and Rhodamine dye degradation.  The manuscript focused primarily on the characterization of the materials and their performance in photocatalytic CO2 reduction, while the supplementary data contained most of the Rhodamine dye degradation data.

Generally, I found the manuscript challenging to read.  I recommend that the manuscript be edited to improve readability.  Make sentences concise and make paragraphs shorter.  Each paragraph should have a leading statement followed by supporting statements.  Make sure that all acronyms are clearly defined.

The authors characterized the materials very well, which makes the study publishable with major revisions.  The revisions that I would like to see in this manuscript include more comprehensive discussions on the material characterizations, which I felt were often lacking in the manuscript.  Describe how data from the characterization methods are used and how they lead to or relate to the photocatalytic activity of the materials.  For many figures, the data was presented but not discussed in sufficient detail to allow the reader to see connections between the characterizations and the photocatalyst performance. 

Q.1. The XRD and FTIR spectra are not impacted at all by the incorporation of DHB?  Does it make sense that there are no observable crystalline or chemical changes in CN when DHB is incorporated into its structure?  Explain.  Maybe using Figure 1 could explain the results?

Reply. Thanks for your nice comments. Dear professor, the process of copolymerization (molecular engineering) alter no any changes in the crystallinity and chemical integrity of CN because the amount of organic monomer DHB incorporated is very small (low) and we clearly mention in the result and discussion portion of XRD and FTIR. We have given appropriate references for justification. For your convenience I am going to highlight as yellow that text. Please have a look. Thank you.

Q.2. The BET adsorption isotherm for the materials is a type 3 adsorption isotherm, which suggests that the adsorbing gas (N2 in this case) has a stronger adherence to itself than with the photocatalyst.  Since one step of the photocatalytic mechanism is adsorption of CO2 onto the surface of the catalyst, I wonder if the adsorption of CO2 could be a limiting step in the photocatalytic reduction of CO2.

Reply. Thanks for your nice comments. Dear professor, the first portion of question is related with BET and having H3 hysteresis type loops for all of our samples suggesting that all samples are mesoporous in nature.

Second step is related with adsorption of CO2 onto the surface of the catalyst. So as the materials employed for CO2 adsorption are the most used inorganic and organic photocatalysts due to their thermal stability and defined pore size. These materials can be used in a pure form, with variations or functionalization in their structure and applied in carbon capture or membrane separation processes. Carbon nitride (CN) is inorganic photocatalyst and we used the process of copolymerization for its modification and the framework of zeolites may vary according to the proportion of organic monomer DHB ratio, resulting in arrangements with varying diameters and different adsorption capacities. As clearly mention that copolymerization process is the integration of organic monomer DHB with inorganic CN photocatalyst and the understanding of acidity and alkalinity of sites available on the adsorbent is of great importance because it allows the understanding of higher or lower solid-gas affinity. In this case, since carbon dioxide is an acid gas, the use or modification of the surface properties of CNU-DHBx, make them more alkaline was sought. In general, the adsorption capacity of a solid is determined by their physical properties, surface availability and pore size selectivity required for mass transfer, but also influenced by their chemical properties and affinity with mention materials. Our results indicate that adsorption on CNU-DHBx demonstrate more improvements as compared of parental CNU and also photocatalytic CO2 reduction improved significantly. As CO2 is a linear molecule and linearly adsorbed onto the CNU-DHBx catalyst, since the adsorbed CO2 on the CNU-DHBx catalyst was activated into CO and H2 species. The CNU-DHBx catalyst showed higher activity and stability under high surface area than the CNU catalyst during the CO2 reduction. This must be closely related to the higher activity of CNU-DHBx for CO2 adsorption and activation on its surface. Thanks again for your great comments.

Q.3. Why is the BET surface area of the composite material a factor of 2 or more higher than that of the parent compound?   Explain the possible reasons why DHB would open up surface areas of the materials without impacting the crystalline structure of chemical structure according to XRD and FTIR.  The authors do point out that various gases are liberated from the materials during the synthesis, which may increase mesoporosity.   

Reply. Thanks for your nice comments. Dear professor, the surface area of our copolymerized materials increased because the incorporation of monomer DHB is so small and so numerous that their surface area is higher than that of the CNU. Generally, surface area decreases because of pore blocking / clogging of the integrated dopant species (large amount).  One can also see that higher the amount of DHB within CNU decrease the surface area is the result of blocking site of main material.

Secondly, large surface area mainly related with thermal pretreatment of the specific surface area analysis. Pure CNU have a low thermal stability and can transform upon oxidation (calcination in air) so the porosity could be greatly affected.

Here we already applied these both condition that

  • Our copolymerized DHD is so small in amount within CNU that increase their surface area
  • I applied the process of co-condensation (calcinations) process for synthesis of our materials.

Thanks again.

Q.4. The hysteresis in this case, as pointed out by the authors, is an H3 type of hysteresis, suggesting that pore structure is irregular and that the pores are characteristic of flake-like granular materials.  This is supported by the SEM pictures presented in the manuscript.

Reply. Thanks for your nice comments. Dear professor, parental CNU having small surface area as compared of modified sample (CNU-DHBx) and we demonstrate its SEM morphology as an agglomerated shape having irregular small stacking flakiness as displayed in (Fig. 4a,b), while this morphology became elongated in size after the inducing of DHB monomer within CNU, having large cloudy superficial area as shown in (Fig. 4c,d).

Secondly, BET surface area is inversely proportional to the pore diameter. This is because of the fact that particles with smaller pores diameter display larger specific surface area. However, porosity is another important parameter defining the alteration in BET surface area. In some cases, particles with small pore diameters shows small surface area due to small number of pores per gram.

Thanks again.

Q.5. The diffuse reflectance UV-Vis data, as mentioned by the authors, show that the DHB-CNU materials absorb light in the visible light spectrum, and therefore, they have lower bandgaps than the CNU material by itself.  Can the authors use this data to determine the bandgaps of the materials and report them in this manuscript?  If so, then please do.  If not, then why not?

Reply. Thanks for your nice comments. Dear professor, we draw a descriptive table in supporting information (Table.S2) in which we manifested band gap of all samples, C/N ratio, surface area and photocatalytic performance of all materials respectively. We also include several sentences in the text to describe the aspect of current table in different places. For your convenience I am going to highlight as yellow. Please have a look. Thank you again.

Q.6. Explain why the open circuit photo response behavior of copolymerized samples were inconsistent with the SPV and PL data.  What does this inconsistency mean?  Was the data reproduced to verify it?

Reply. Dear professor, thanks for pointing out this mistake. We removed the above-mentioned sentence in revised manuscript. Thank you again. Please have a look.

Q.7. Can you hypothesize why there is an optimum amount of DHB in the composite material, as shown in Figure 7b, using the photocatalytic mechanism as a basis?

Reply. Thanks for your nice comments. Dear professor, in figure. 7b we utilized all of our synthesized samples for photocatalytic conversion of CO2 into binary source of CO and H2. The purpose of these experiments is only the comparison that 15 mg of DHB monomer is the optimum amount to be incorporated within CNU, increase surface area, reduced band gap, faster photoinduced electrons and holes, reduced charge recombination, optimized the optical absorption, hence increase remarkable the photocatalytic performance of CNU respectively.  Thanks again for your comments.

Q.8. In the time-resolved photoluminescence data (7e), it appears that the exciton lifetime of the CNU-DHBx composite is shorter than that of the CNU by itself.  This seems to suggest that recombination of electron-hole pairs would be more significant in the composite than in the CNU by itself.  Is this data contradictory of other characterization data?  Why or why not? 

Reply. Thanks for your nice comments. Dear professor, it is a blinder mistake, and we modified Figure.7e in revised manuscript. Thanks again

Q.9. When the authors state that the photocatalyst is efficient in photoreduction of CO2, please define efficiency in the context of the manuscript. 

Reply. Thanks for your nice comments. Dear professor, the meaning of efficient mean remarkable, superior. Because after copolymerization of DHB, our superior sample CNU-DHB15.0 manifesting a momentous photocatalytic activity as comparison and therefore it is an efficient photocatalyst for the photoreduction of CO2 into CO and H2. For more explanation, please see the Table.S2 in supporting information and results and discussion of photocatalytic performance along with mechanism. For your convenience I am going to highlight as yellow. Please have a look. Thank you again.

Q.10. Table S3 summarizes many efforts in photocatalytic CO2 reduction by incorporating various monomers in urea CN, and so this effort is a variation on a theme of research efforts conducted by one or more of the authors of this manuscript.  Therefore, I would not use the word "novel", since this idea has been published multiple times using other monomers incorporated into CNU. The DHB incorporated into CNU in this study, however, apparently has relatively high photoactivity for CO2 reduction compared to many of the other materials tried by this research group, according to Table.S3. 

Reply. Thanks for your nice comments. Dear professor, Table.S1 demonstrate the reported work with our present work, in which our photocatalytic efficiency is more superior than all of reported work. Secondly, I already study these reported papers in which I used different parameters as compared on level of solvents, sacrificial agent, co-catalyst, quality and quantity of catalyst and irradiation wavelength and time. For your convenience I am going to highlight as yellow. Please have a look. Thank you again.

Reviewer 3 Report

In this manuscript, the significant enhancement of photocatalytic activity of carbon nitride incorporated with dihydroxy benzene was
revealed, what could be used for CO2 reduction.
The results are reliable and the investigation is interesting. The manuscript should be slightly revised before publication.

page 2
line 56
... or less potentials towards photocorrosion. 
line 65
... has almost low surface areas and less active sites, 
line 68
causing bottleneck in the efficiency of CN 
line 71
sphere CN

page 3
line 108
cop-polymerization 

page 12
Figure 9
Authors probably mean that water gets oxidized to oxygen at holes sites.

Author Response

Reviewer:2.

Comments and Suggestions for Authors

In this manuscript, the significant enhancement of photocatalytic activity of carbon nitride incorporated with dihydroxy benzene was revealed, what could be used for CO2 reduction. The results are reliable and the investigation is interesting. The manuscript should be slightly revised before publication.

Q.1. page 2. line 56... or less potentials towards photocorrosion.

Reply. Thanks for your experience comments. Dear Professor, we did correction in revised manuscript. The details are as follows.

Unfortunately, such semiconductor materials have a broad bandgap that absorbs only ultraviolet light irradiation and has less photocorrosion protentional.

Q.2. line 65... has almost low surface areas and less active sites,

Reply. Thanks for your experience comments. Dear Professor, we did correction in revised manuscript. The details are as follows

Synthesis of bulk CN By the direct co-condensation process limit their applications because of low surface areas and less active cite.

Q.3. line 68 …. causing bottleneck in the efficiency of CN

Reply. Reply. Thanks for your experience comments. Dear Professor, we did correction in revised manuscript. The details are as follows

To improve quantum efficiency in the visible light region, the above-mentioned drawbacks must be overcome that limit the performance of CN.  

Q.4. line 71 …… sphere CN

Thanks for your experience comments. Dear Professor, we did correction in revised manuscript. The details are as follows

Micro/nanostructured CN was obtained using a variety of techniques, including hard templating, which was commonly used to obtain porous, sphere, and tubular carbon nitride (CN).

Q.5. page 3 ……..line 108

Reply. Thanks for your experience comments. Dear Professor, we correct the word copolymerization in revised manuscript.

Q.6.Figure 9. Authors probably mean that water gets oxidized to oxygen at holes’ sites.

Reply. Thanks for your experience comments. Dear Professor, in photocatalysis have two reaction step that one is the photogenerated electrons in the conduction band of materials that here resulted as for photocatalytic CO2 reduction. Second step is the photogenerated holes formation due to oxidation process and resulted in the photodegradation of RhB pollutants respectively. The attachment of oxygen with water create space on the surface of catalyst in the ground floor (VB) that absorb dye during degradation. Thanks again.

Round 2

Reviewer 1 Report

Accepted for publications.